# A versatile workflow to integrate RNA-seq genomic and transcriptomic data into mechanistic models of signaling pathways

**Martín Garrido-Rodriguez**[1,2,3,4], **Daniel Lopez-Lopez**[1,5], **Francisco M. Ortuno**[1,5], **María Peña-Chilet**[1,5,6], **Eduardo Muñoz**[2,3,4], **Marco A. Calzado**[2,3,4], **Joaquin Dopazo**[1,5,6,7]*

1 Clinical Bioinformatics Area, Fundación Progreso y Salud (FPS), Hospital Virgen del Rocío, Sevilla, Spain, 2 Departamento de Biología Celular, Fisiología e Inmunología, Universidad de Córdoba, Córdoba, Spain, 3 Instituto Maimónides de Investigación Biomédica de Córdoba (IMIBIC), Córdoba, Spain, 4 Hospital Universitario Reina Sofía, Córdoba, Spain, 5 Computational Systems Medicine, Institute of Biomedicine of Seville (IBIS), Sevilla, Spain, 6 Centro de Investigación Biomédica en Red de Enfermedades Raras (CIBERER), FPS, Hospital Virgen del Rocío, Sevilla, Spain, 7 FPS/ELIXIR-es, Hospital Virgen del Rocío, Sevilla, Spain

* joaquin.dopazo@juntadeandalucia.es

**Data Availability Statement:** MIGNON is available from https://github.com/babelomics/MIGNON and its documentation can be found at https://

## Abstract

MIGNON is a workflow for the analysis of RNA-Seq experiments, which not only efficiently manages the estimation of gene expression levels from raw sequencing reads, but also calls genomic variants present in the transcripts analyzed. Moreover, this is the first workflow that provides a framework for the integration of transcriptomic and genomic data based on a mechanistic model of signaling pathway activities that allows a detailed biological interpretation of the results, including a comprehensive functional profiling of cell activity. MIGNON covers the whole process, from reads to signaling circuit activity estimations, using state-of-the-art tools, it is easy to use and it is deployable in different computational environments, allowing an optimized use of the resources available.

## Author summary

Currently, RNA massive sequencing RNA-seq is the most extensively used technique for gene expression profiling in a single assay. The output of RNA-seq experiments contains millions of sequences, generated from cDNA libraries produced by the retro-transcription of RNA samples, that need to be processed by computational methods to be transformed into meaningful biological information. Thus, a number of bioinformatic workflows and pipelines have been proposed to produce different types of gene expression measurements, including in some cases, functional annotations to facilitate biological interpretation. While most pipelines focus exclusively on transcriptional data, the ultimate activity of the resulting gene product also depends critically on its integrity. Although traditional hybridization-based transcriptomics methodologies (microarrays) miss this information, RNA-seq data also contains information on variants present in the transcripts that can affect the function of the gene product, which is systematically ignored by current RNA-

babelomics.github.io/MIGNON/. Additionally, we have prepared a bash script to perform a dry run. The instructions can be found at https://babelomics.github.io/MIGNON/1_installation.html. The data used in the examples and figures of this manuscript is freely available at: https://figshare.com/articles/dataset/MIGNON_data/13286627/1.

**Funding:** JD has received these grants: SAF2017-88908-R from the Ministerio de Economía y Competitividad and PT17/0009/0006 from the Instituto de Salud Carlos III, as well as an FP7 People Marie-Curie Actions 813533 and and Horizon 2020 Framework Programme 676559. The funders had no role in study design, data collection and analysis, decision to publish, or preparation of the manuscript.

**Competing interests:** The authors have declared that no competing interests exist.

seq pipelines. MIGNON is the first workflow able to perform an integrative analysis of transcriptomic and genomic data in the proper functional context, provided by a mechanistic model of signaling pathway activity, making thus the most of the information contained in RNA-Seq data. MIGNON is easy to use and to deploy and may become a valuable asset in fields such as personalized medicine.

This is a *PLOS Computational Biology* Software paper.

## Introduction

Because of the plummeting in the cost of sequencing technologies during the last decade, RNA massive sequencing (RNA-seq) has become mainstream to study the transcriptome [1]. Currently, short-read sequencing technologies, typically producing outputs of 30 million reads per sample, are the most extensively used methodologies for gene expression profiling [2]. This pace of data generation requires computational processing to produce interpretable results. Thus, the use of pipelines to perform the different steps of transcriptomic data processing have become a widespread practice. The core of these is usually composed by spliced aligners as **STAR** [3], **HISAT2** [4] or **Rail-RNA** [5], which map reads against a reference genome, or by pseudo-alignment tools as **Salmon** [6] or **Kallisto** [7], that directly obtain a quantification for the regions of interest using probabilistic models. Additionally, there are pipelines which are intended to be run by the user in local computers or high-performance environments, as **QuickRNASeq** [8], or interactively in cloud-based platforms, after uploading raw data to an external service, as **BioJupies** [9] or **RaNA-Seq** [10]. Typically, the interpretation of the experiment involves differential expression analysis, carried out using count based or linear models, with packages as **edgeR** [11], **DESeq2** [12] or **limma** [13], followed by methods, such as over representation analysis [14] or the gene set enrichment analysis [15], to extract functional information from the obtained results.

Despite different pipelines to perform the aforementioned tasks are available (Tables 1 and 2), most of them present two major drawbacks. First, the genomic information contained in the RNA-Seq reads usually remains unused. However, genomic variants, which may contain crucial information about the functionality and potential activity of the resulting proteins in the different processes where they participate, can be retrieved from such sequences. In this sense, it is well known that RNA-Seq has some limitations for DNA variant calling. There are two main points to consider: (i) lowly expressed genes include lower depth, so variant calling is harder in those regions and (ii) the detection of heterozygous variants can be limited due to allele-specific gene expression [16]. Despite these limitations, it has been demonstrated that variants can be called even for low expressed genes in deeper RNA-Seq sequence samples. Moreover, some studies have shown that RNA-Seq variant calling is able to provide a good sensitivity of 99.7%-99.8% in both heterozygous and homozygous variants whereas precision still reaches 97.6% in homozygous but 90% in heterozygous [17]. The second major drawback is that conventional functional analysis strategies are mainly descriptive, and very limited in providing biological insights of the underlying molecular mechanisms that produce the observed phenotypic responses. Recently, a new generation of methods, known as mechanistic pathway analyses, are outperforming traditional approaches in both biological explanatory power and interpretability [18]. Here we present MIGNON (Mechanistic InteGrative aNalysis

**Table 1. Features of the workflows for RNA-seq data analysis.**

| Workflow | URL | Google scholar citations | Year | Implementation | Read pre-processing | (Pseudo) Alignment | Variant calling and annotation | Differential gene expression | Functional analysis | Omic integration |
|---|---|---|---|---|---|---|---|---|---|---|
| QuickRNASeq | https://sourceforge.net/projects/quickrnaseq/ | 26 | 2016 | Shell, Perl and R scripts | - | STAR | VarScan2 | - | - | No |
| SePIA | http://anduril.org/sepia | 25 | 2016 | Anduril workflow | FastX-Toolkit Trimmomatic TrimGalore | STAR TopHat Bowtie Bowtie2 | Bambino ANNOVAR | Cuffdiff DESeq DESeq2 EdgeR | SPIA | No |
| Recount2 | https://jhubiostatistics.shinyapps.io/recount/ | 154 | 2017 | Shiny app and R package | - | Rail-RNA | - | - | - | No |
| RNACocktail | https://bioinform.github.io/rnacocktail/ | 71 | 2017 | Python scripts | - | HISAT2 | GATK | DESeq2 | - | No |
| BioJupies (ARCHS4) | https://amp.pharm.mssm.edu/biojupies/ | 143 | 2018 | Web service | - | Kallisto | - | Limma Characteristic direction | Enricher | No |
| GREIN | https://shiny.ilincs.org/grein | 9 | 2019 | Shiny app and R package | Trimmomatic | Salmon | - | EdgeR | - | No |
| VaP | https://modupeore.github.io/VAP/ | 1 | 2019 | Perl scripts | AfterQC Trimmomatic | TopHat2 HISAT2 STAR | GATK ANNOVAR VEP | - | - | No |
| DEWE | http://www.sing-group.org/dewe/ | 3 | 2019 | Java app | Trimmomatic | Bowtie2 HISAT2 | - | Ballgown EdgeR | PathfindR | No |
| RaNa-Seq | https://ranaseq.eu/ | 0 | 2019 | Web service | Fastp | Salmon | - | DESeq2 | GOseq fgsea | No |
| **MIGNON** | https://github.com/babelomics/MIGNON | - | **2020** | **WDL workflow** | **Fastp** | **Salmon STAR HISAT2** | **GATK VEP** | **EdgeR** | **hiPathia** | **Yes** |

Of rNa-seq), a complete and versatile workflow able to exploit all the information contained in RNA-Seq data and producing not only the conventional normalized gene expression matrix, but also an annotated VCF file per sample with the corresponding mutational profile.

**Table 2. Analysis outputs of the workflows for RNA-seq data analysis.**

| Workflow | Normalized gene expression | Differential gene expression | Transcriptomic-based functional results | Genomic variants | Annotated genomic variants | Integrated Transcriptomic + genomic functional results |
|---|---|---|---|---|---|---|
| QuickRNASeq | Yes | - | - | Yes | - | - |
| SePIA | Yes | Yes | Yes | Yes | Yes | - |
| Recount2 | Yes | - | - | - | - | - |
| RNACocktail | Yes | Yes | - | Yes | - | - |
| BioJupies (ARCHS4) | Yes | Yes | Yes | - | - | - |
| GREIN | Yes | Yes | - | - | - | - |
| VaP | - | - | - | Yes | Yes | - |
| DEWE | Yes | Yes | Yes | - | - | - |
| RaNa-Seq | Yes | Yes | Yes | - | - | - |
| **MIGNON** | **Yes** | **Yes** | **Yes** | **Yes** | **Yes** | **Yes** |

Moreover, MIGNON can combine both files to model signaling pathway activities through an integrative functional analysis using the mechanistic modeling algorithm Hipathia [19]. Signaling circuit outputs can further be easily linked to phenotypic features (e.g. disease outcome, drug response, etc.) [19–21]. Mechanistic modeling has been successfully applied to understand disease mechanisms in rare diseases [22,23], complex diseases [21], and, especially in cancer [19,24–26]. Specifically, the hiPathia algorithm has demonstrated to have a superior sensitivity and specificity than other similar algorithms available [27].

## Design and implementation

### Workflow implementation

The complete pipeline was developed using the Workflow Description Language (WDL, https://github.com/openwdl/wdl) due to its flexibility, human-readability and easy deployment. Thus, all the steps of the pipeline were wrapped into WDL tasks that were designed to be executed on an independent unit of containerized software through the use of docker containers, which prevent deployment issues using an independent environment for each execution. The workflow can be executed in personal computers or in high-performance computing (HPC) environments, both locally or in cloud-based services with **cromwell** (https://github.com/broadinstitute/cromwell), a Java based software that control and interpret WDL, using a JSON file as input. To run MIGNON, three dependencies are required: Java (v1.8.0), cromwell and an engine able to run the containerized software (i.e Docker or Singularity). The list of docker containers employed by MIGNON can be found at the S1 Table.

### Quality control and alignment

First, using raw sequences as the input for the workflow, **fastp** (v0.20.0) [28] is applied to perform the quality trimming and filtering of reads using the default values for windows size and required mean quality and length. Then, **FastQC** (v0.11.5) can be used to create a quality report for each pre-processed read file. After the quality control step, five modes for the execution of the workflow can be selected (see Table 3). Each execution mode uses a different combination of "core" tools to perform the alignment or pseudo-alignment of pre-processed reads, as explained in the tool documentation (see also Fig 1). In brief, all of them make use of a combination of **STAR** (v2.7.2b), **HISAT2** (v2.1.0), **Salmon** (v0.13.0) and **FeatureCounts** (v1.6.4) [29] to align (or pseudo-align) reads against a reference genome (or transcriptome) and subsequently obtain the counts per gene matrix. The *hisat2* and *star* modes use a conventional counting strategy, employing **FeatureCounts** to summarize the number of sequences overlapping the genomic regions of interest (genes), as specified by a genome annotation file. On the other hand, the core component of *salmon-hisat2*, *salmon-star* and *salmon* consist of the pseudo-aligner **Salmon**, which directly obtains transcript level quantification using a probabilistic model. Note that in the *salmon-hisat2* and *salmon-star* modes, the execution of **STAR** or

**Table 3. MIGNON execution modes.**

| Execution mode | Alignment | Quantification | Variant calling | Computational profile |
|---|---|---|---|---|
| *salmon-hisat2* | HISAT2 | Salmon | Yes | Low memory consumption. Slower than STAR. |
| *salmon-star* | STAR | Salmon | Yes | High memory consumption. Faster than HISAT2. |
| *hisat2* | HISAT2 | featureCounts | Yes | Low memory consumption. Slower than STAR. |
| *star* | STAR | featureCounts | Yes | High memory consumption. Faster than HISAT2. |
| *salmon* | - | Salmon | No | Low memory consumption and fast. |

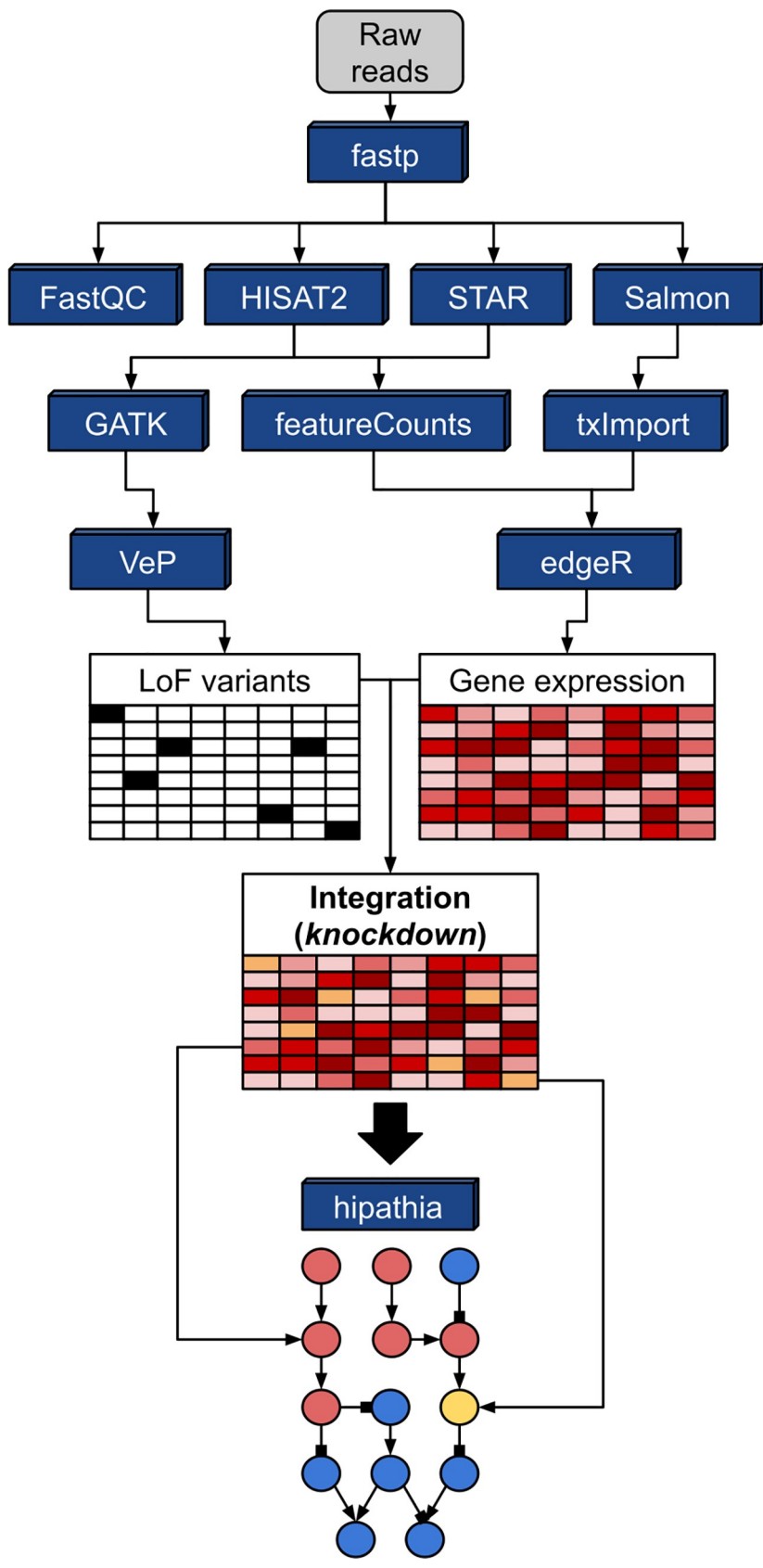

**Fig 1. MIGNON Workflow.** Directed graph summarizing the tools employed by the workflow (blue boxes) and the strategy used by MIGNON to integrate genomic and transcriptomic information into signaling circuits. Gene expression and LoF variants are obtained from reads and integrated by doing an in-silico knockdown of genes that present a LoF variant. Then, this combined matrix is used as the input for hiPathia, that estimates the signaling circuits activation status by using expression values as proxies for protein signaling activities.

**HISAT2** is still necessary to generate the alignment files that feed the variant calling sub-workflow.

## Variant calling and annotation

Genomic data for the expressed genes can be inferred from reads through variant calling. Due to the number of intermediate steps carried out during this process, it was encapsulated on an independent sub-workflow which is run at sample level. On it, the input material consists of the alignments generated with STAR or HISAT2 and the output is a list of variants in the variant call format (VCF). The whole process is performed using the Genome Analysis ToolKit (v4.1.3.0) [30], and it was designed following the GATK best practices for the variant calling from RNA-Seq data. Similar to germline variant discovery with DNA sequencing, this sub-workflow specifically includes a step to mark duplicate reads, which will help to reduce the direct dependency of the depth by gene expression. Additionally, the pipeline also includes other steps to specially deal with RNA-Seq peculiarities for variant calling. Thus, some aligned reads are reformatted in order to control the expansion produced by introns. Specifically, reads are split into separate reads when introns are identified inside, thus reducing artifacts in the downstream variant calling. Mapping qualities are also reassigned and adapted to match DNA conventions. Finally, in order to avoid variants called under low evidence, our sub-workflow includes a filter by depth step to only keep those variants found in at least a number of reads (by default $>5$) as recommended in the literature [16]. The output VCFs are then annotated with the Variant Effect Predictor (VeP v99) [31], a powerful tool for the prioritization of genomic variants that summarizes in two scores (Polyphen [32] and SIFT [33]) the predicted impact of variants on protein stability and functionality.

## Normalization and differential expression analysis

The different execution modes converge at the counts per sample matrix, which is the output of **FeatureCounts**. On the other hand, for **Salmon** quantifications, the count matrix is generated with **txImport** (v1.10.0) [34] and a transcript-to-gene file. The *lengthScaledTPM* option is used to correct the estimated counts by both transcript length and library size. Then, RNA-seq gene level counts are normalized with the Trimmed mean of M values (TMM) method and conventional differential gene expression analysis can be performed with the **edgeR** package (v3.28.0) [11].

## Integrative mechanistic signaling pathway activity analysis

The HiPathia R package (v2.2.0) [19] is used to perform the functional analysis, either using transcriptomic data alone, or integrating them with the genomic data. HiPathia implements a mechanistic model of signaling pathways that, using gene expression values as proxies of protein activities, infer signaling circuit activities and the corresponding functional profiles triggered by them. Since the model is mechanistic, it allows to infer the effect of an intervention (e.g., a knock-out) on the resulting signaling (and functional) profile [35], a concept that can easily be assimilated to a loss of function (LoF) [21]. In practical terms, MIGNON considers that a gene harbors a LoF if it presents at least one variant with a SIFT score $< 0.05$ and a

PolyPhen score $> 0.95$ (default values that can be modified by the user). Then, an *in-silico* knock-down is simulated by multiplying the scaled normalized expression values by 0.01 only in the affected samples. Next, the HiPathia signal propagation algorithm is applied to obtain the signaling circuit activities. Finally, the profiles of signaling activities of the samples belonging to the groups of interest are compared using a Wilcoxon signed rank test. For more information about the HiPathia method, please refer to [19] and [21].

## Modularity of the workflow

The choice of methods for the different steps of MIGNON was based on two recent benchmarking evaluations of the processes to perform the primary analysis of RNA-seq data [1,36]. However, the modular design of the pipeline makes it easy to replace any tool for another one providing it matches the input/output schema used. Thus, users can easily replace tools in the pipeline by making small changes to the MIGNON WDL code, as explained in the documentation (https://babelomics.github.io/MIGNON/4_advanced.html#modularity).

## Results

### MIGNON integrative approach for the mechanistic interpretation of multi-omic information into a pathway framework

MIGNON is the first pipeline able to extract genomic and transcriptomic information from RNA-seq data and integrate them within a mechanistic framework. The ultimate protein activity is assessed from the transcriptional activity conditioned to the integrity of the gene. No matter its level of expression, a gene that harbors a deleterious mutation is *in-silico* knocked-down by the model to simulate the loss of function (**Fig 1**). To evaluate how the proposed strategy affects the predicted signaling circuit activities, two different runs of MIGNON were carried out over 462 unrelated human lymphoblastoid cell line samples from the 1000 Genomes sample collection, corresponding to the CEU, FIN, GBR, TSI and YRI populations [37]. In the reference run, only transcriptomic information (raw) was used, while in the case example run the knock-down strategy was applied. **Fig 2A and 2B** clearly depicts how the knock-down due to LoF mutations interrupts the transduction of the signal in three circuit/sample pairs. Moreover, **Fig 2C** shows that the overall predicted signaling circuit activities are significantly lower (paired Wilcoxon signed-rank test P value $< 2.2\text{x}10^{-16}$) when the genomic information is integrated in the model. This example clearly shows how the use of transcriptomics data alone produced an incomplete picture of the real signaling activity and proves the usefulness of multi-omic data integration.

## Workflow performance evaluation

To assess MIGNON performance and resource consumption, the workflow was executed over 6 different human datasets (S2 Table), comprising a total of 42 samples. It was tested with cromwell (v47) and singularity (v3.5), using 6 different CPU configurations on tasks allowing multi-threading. This analysis revealed that the slower components of the workflow are the aligners (HISAT2 and STAR) and the *MarkDuplicates* and *HaplotypeCaller* steps of the GATK sub-workflow. **Fig 3** summarizes the time and memory consumption of the tools which allow multi-threading using 6 different CPU configurations. While HISAT2 is slower than STAR, the second one makes a more intensive use of available memory. Therefore, both aligners are available in MIGNON since this tradeoff should be considered if planning to deploy the workflow in cloud-computing based environments or, contrarily, in limited memory computing environments. Additionally, **Fig 4** shows the time and memory consumption of the different

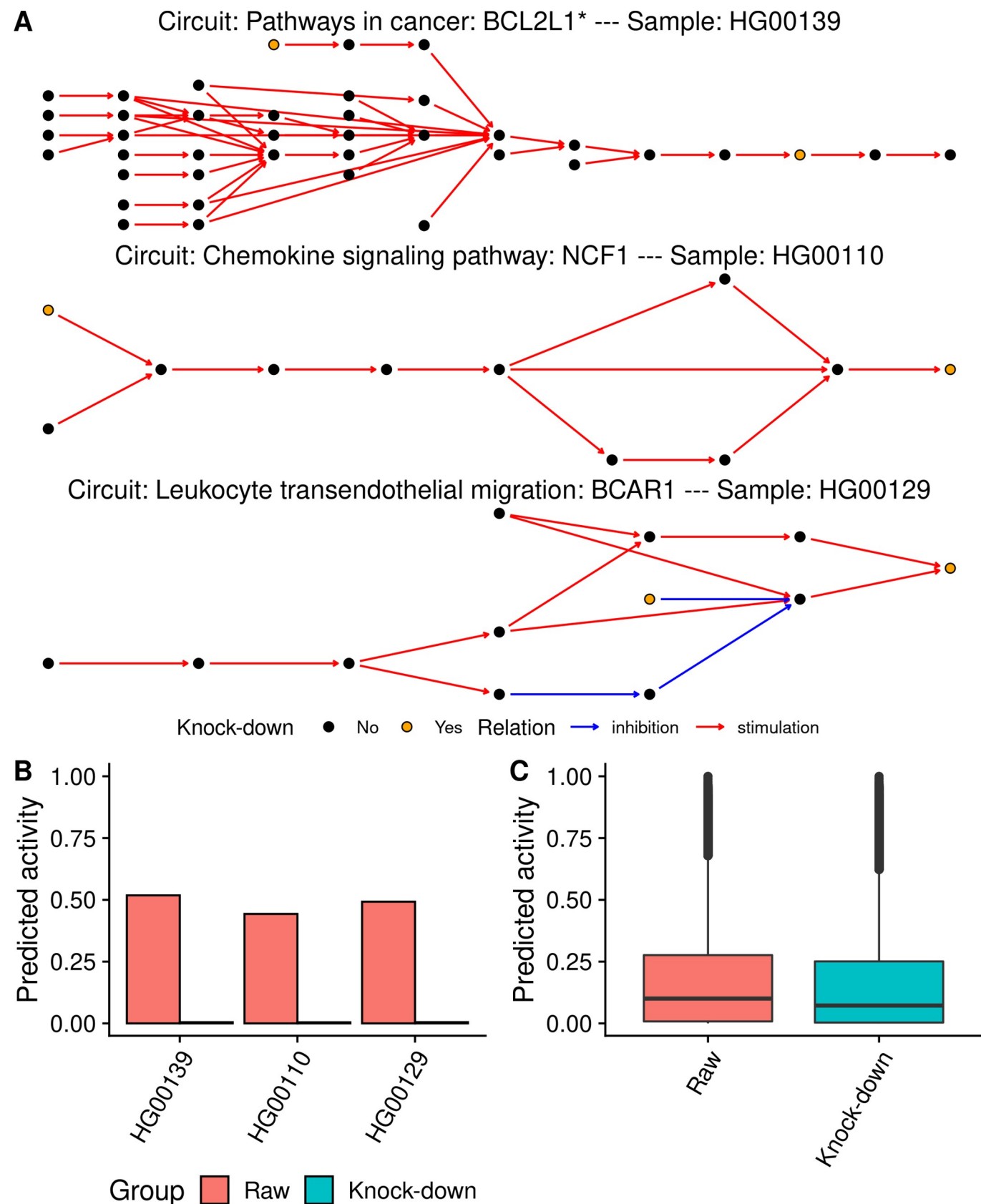

**Fig 2. *In silico* knock-downs effect on predicted signaling circuit activities.** A) Network representation of three signaling circuits that contain genes with loss of function variants for three subjects from the 1000 genomes cohort. The node color indicates whether a gene contained in it has a loss of function variant (yellow) or not (black). Red and blue arrows indicate stimulations and inhibitions, respectively. B) Predicted signaling activity for three circuit/sample pairs on the sub-figure A. Color represents signaling circuit activity with and without considering the genomic information. C) Violin plots showing all the predicted signaling circuit activities with and without the genomic information for the 1000 genomes cohort (paired Wilcoxon signed-rank test P value < 2.2x10$^{-16}$).

steps that compose the GATK sub-workflow. Here, *MarkDuplicates* displays the highest memory consumption and *HaplotypeCaller* shows the longest runtime. Overall, the different tasks carried out by the workflow show a maximum memory usage under the 32 gigabytes, which makes the pipeline deployable under most computational environments. Finally, and due to the WDL, cromwell and docker combination, the workflow is something fast and easy to deploy and setup.

## Functionality of current available workflows

In order to have a comprehensive list of available pipelines for RNA-seq data processing, only those published from 2015 onwards and able to use raw read files (fastq) as input data were considered. Nine workflows fulfilled these criteria: QuickRNASeq [8], SEPIA [38], Recount2 [39], RNACocktail [36], ARCHS4 [40], GREIN [41], VaP [17], DEWE [42] and RaNA-Seq [10]. Table 1 list the components implemented in each pipeline. Since their performances depend on their components, which are similar across them, a comparison of their respective functionalities is listed in Table 2. The first noticeable aspect is that, although some of them can carry out variant calling (QuickRNASeq, SEPIA, RNACocktail and VaP), none of them

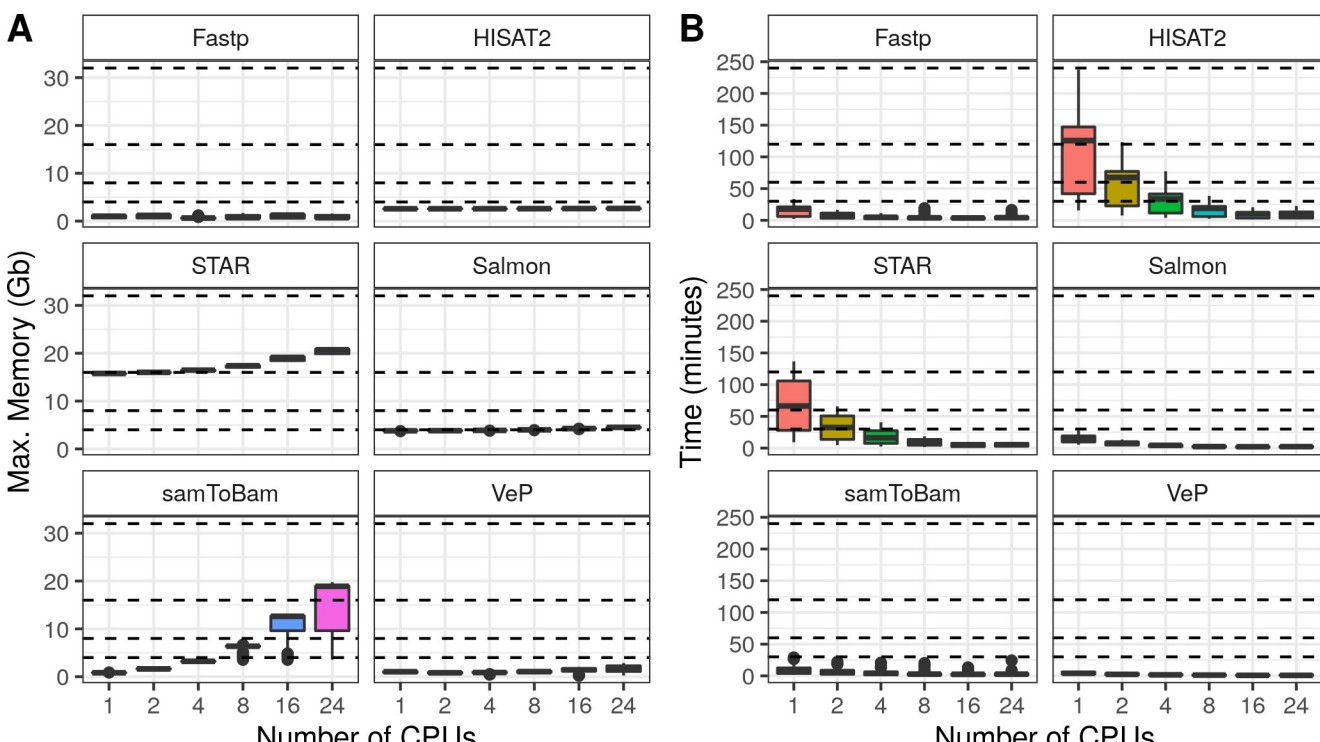

**Fig 3. MIGNON performance results.** Multi-thread tasks. **A**) Memory consumption by task. Each boxplot represents the maximum memory consumption in Gigabytes (y axis) for each CPU configuration (X axis) and each multi-thread task (facets). Dashed lines indicate the following memory configurations: 4, 8, 16 and 32 gigabytes. **B**) Elapsed time by task. Each boxplot represents the elapsed time (Y axis) for each CPU configuration (X axis) and each task (facets). Dashed lines indicate time points: 30, 60, 120 and 240 minutes.

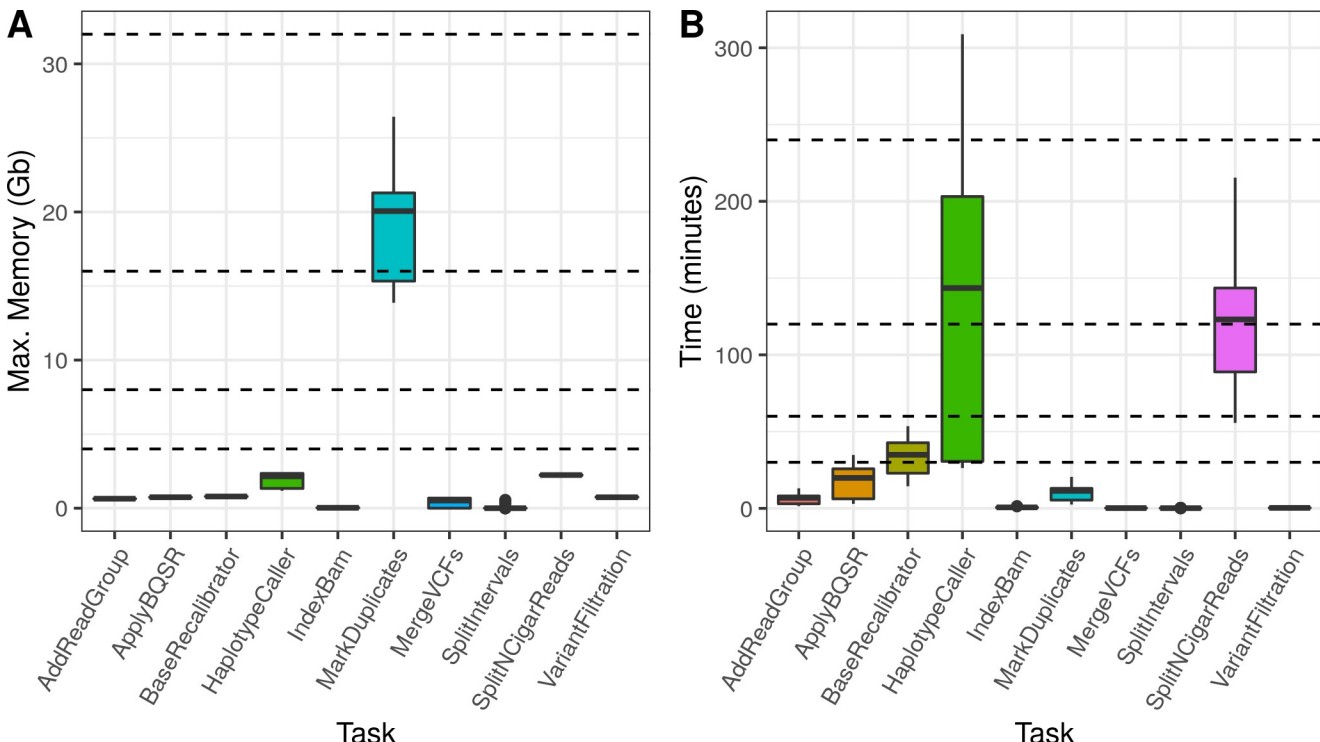

**Fig 4. GATK sub-workflow performance results. A**) Memory consumption by task. Each boxplot represents the maximum memory consumption in Gigabytes (Y axis) for each task (X axis). Dashed lines indicate the following memory configurations: 4, 8, 16 and 32 gigabytes. **B**) Elapsed time by task. Each boxplot represents the elapsed time (Y axis) for each task (X axis). Dashed lines indicate the following time points: 30, 60, 120 and 240 minutes.

provides a way to integrate the called variants with the gene expression data as MIGNON does. Among the workflows, only SEPIA provides an option for functional analysis of both omic results (obviously transcriptomic and genomic data are interpreted independently). Although the real usage level of these workflows is always difficult to estimate, Google Scholar citations can provide an approximate measurement of the relative impacts in terms of scientific document quotations. According to these observations, SEPIA displays a modest 6% of use among the available workflows. Conversely, Recount2 (36%), ARCHS4 (33%) and RNA-Cocktail (16%) together account for 85% of the citations. Among these, only one (ARCHS4) provides functional analysis, by conventional enrichment analysis. Thus, a workflow capable, not only to extract transcriptomic and genomic information from RNA-seq reads, but also to integrate them and to provide a functional analysis in a sophisticated framework of mechanistic modeling of signaling pathways seems to be a good step forward.

## Conclusions

In summary, MIGNON represents an innovative concept of RNA-Seq data analysis that automates the sequence of steps that leads from the uninformative raw reads to the ultimate sophisticated functional interpretation of the experiment, providing, for the first time, a user-friendly framework for integration of genomic and transcriptomic data.

MIGNON makes use of several popular methods to perform the initial processing of reads and utilize the HiPathia mathematical model to provide a mechanistic interpretation of the experiment in the context of human signaling. MIGNON has an enormous application potential in personalized medicine, especially in the analysis of cancer transcriptomes, given its

ability to interpret putative driver mutations along with gene expression in the context of signaling activity, a process highly relevant in tumorigenesis.

MIGNON can be easily deployed in different computer environments making an optimal use of the resources. Additionally, the modularity with which the workflow has been designed makes its upgrade and maintenance a straightforward task.

## Supporting information

**S1 Table. List of docker containers employed by MIGNON.**
(XLSX)

**S2 Table. List of datasets used to assess MIGNON performance.**
(XLSX)

## Author Contributions

**Conceptualization:** María Peña-Chilet, Eduardo Muñoz, Marco A. Calzado, Joaquin Dopazo.

**Funding acquisition:** Joaquin Dopazo.

**Investigation:** Joaquin Dopazo.

**Methodology:** Martín Garrido-Rodriguez, Francisco M. Ortuno, María Peña-Chilet.

**Software:** Martín Garrido-Rodriguez, Daniel Lopez-Lopez, Francisco M. Ortuno.

**Supervision:** Marco A. Calzado, Joaquin Dopazo.

**Writing – original draft:** Martín Garrido-Rodriguez, Joaquin Dopazo.

**Writing – review & editing:** Joaquin Dopazo.

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
