## [Decision Letter · Decision Letter 0]

9 Oct 2020

Dear Dr Dopazo,

Thank you very much for submitting your manuscript "A versatile workflow to integrate RNA-seq genomic and transcriptomic data into mechanistic models of signaling pathways" for consideration at PLOS Computational Biology.

As with all papers reviewed by the journal, your manuscript was reviewed by members of the editorial board and by several independent reviewers. In light of the reviews (below this email), we would like to invite the resubmission of a significantly-revised version that takes into account the reviewers' comments.

We cannot make any decision about publication until we have seen the revised manuscript and your response to the reviewers' comments. Your revised manuscript is also likely to be sent to reviewers for further evaluation.

Sincerely,

Mihaela Pertea

Software Editor

PLOS Computational Biology

Mihaela Pertea

Software Editor

PLOS Computational Biology

Reviewer's Responses to Questions

**Comments to the Authors:**

Reviewer #1: The manuscript by Garrido-Rodriguez et al., entitled "A versatile workflow to integrate RNA-seq genomic and transcriptomic data into mechanistic models of signaling pathways" provides a novel comprehensive -omics analysis workflow based on containers.

Compared to earlier efforts, novel improvements include the detection and utilization of genomic variances in the RNA-seq data as well as functional annotation for mechanistic pathway analysis. As such, it is potentially of great interest to a wider readership.

The workflow is modeled using the Workflow Description Language, and employs fastp for quality trimming and filtering of reads, FastQC for quality control of read files, as well as STAR, HISAT2, Salmon, FeatureCounts, txImport and edgeR for RNA-seq expression analysis. Normalization applies the standard and recommended Trimmed Means of M-values approach. The Genome Analysis ToolKit and Variant Effect Predictor are used for variant calling and annotation, and HiPathia for functional analysis. Each tool is provided to the user as containerized software via Docker.

While the components are all based on well established methods, the benefits of the proposed approach beyond compiling a range of results -- in a truly integrated analysis -- are not demonstrated in this Research Article. Figures only cover technical issues like runtime and memory use. The authors do not even demonstrate that the claimed lower resource footprint allows the analysis of more demanding sample cohorts.

Seeing that the article already provides an overview of earlier well established workflow systems, the key unanswered question must surely be how the new workflow system compares to these in terms of analysis outcomes rather than "workflow system features".

The direct benefits of the presented workflow to others thus remains rather unclear. To better address this, considering that the authors provide no to little justification for their choice of methods, the authors may in revision choose to extend on this and/or provide a demonstration of how easy it would be to exchange a method in the workflow by another.

While the authors do provide a code repository on github that includes an example script, the provided online documentation is rather basic. The lack of a more comprehensive tutorial or reference limit how easily others could use or build on the presented software.

Minor: Please revise the manuscript for typos that sometimes make the narrative harder to follow.

Overall, while the reported work is in principle of interest in the field, a number of issues would need to be convincingly addressed to make the manuscript of value in a high-profile journal catering to a wider audience. I encourage the authors to resolve these issues in revision.

Reviewer #2: Review of: A versatile workflow to integrate RNA-seq genomic and transcriptomic data into mechanistic models of signaling pathways

The workflow described in the manuscript is an assembly of widely used alignment algorithms and supporting tools for RNA-seq alignment. The pipelines is implemented using the workflow language WDL and utilized the Cromwell engine for execution.

Additional to RNA-seq alignment, the pipeline templates support variant calling.

Comments:

The idea of applying variant calling on RNA-seq data is interesting and has not really been utilized broadly. Providing a solution that works without a long software setup is very helpful since several tools need to be applied in series.

Calling variants reliably from RNA-seq is not commonly done. This is mainly due to the shallow coverage in most RNA-seq experiments compared to exon sequencing. Furthermore, the depth of reads is directly dependent on the gene expression. What can a user expect in terms of quality of called variants? While a full analysis goes probably too far for this publication, the authors should mention existing studies that analyze this issue and briefly discuss them.

The main contribution of the paper is the example scripts found in the Github repository. While they are mostly self-explanatory, it would be good to discuss each of them more clearly in the help section.

When running the example scripts, I ran into several issues related to software version differences compared to the author system setup. For example, the command "gzip -k" command is not supported in many UNIX distributions. Since they are important for system maintenance, they are not easily replaced. (For example, an EC2 instance from AWS does not support this command requiring rewriting).

Since the authors are already relying on Dockerized tools, it would be much better to generate Docker images of the example workflows. It should make the execution of the workflow much more seamless.

I also encountered issues with the curl command to download files. This resulted in corrupted files that had to be manually deleted. (I had to switch to wget to make it work, I don't know why). Maybe file downloads can be validated with a checksum if they are provided?

The aligners can usually come with a relatively long set of parameters during the index generation step or the alignment step as well, such as k-mer length. I assume the parameters are all default. Is there a way to modify them in the Cromwell JSON file?

**Have all data underlying the figures and results presented in the manuscript been provided?**

Reviewer #1: Yes

Reviewer #2: Yes

PLOS authors have the option to publish the peer review history of their article (what does this mean?). If published, this will include your full peer review and any attached files.

Reviewer #1: No

Reviewer #2: **Yes: **Alexander Lachmann
---

## [Decision Letter · Decision Letter 1]

30 Jan 2021

Dear Dr Dopazo,

We are pleased to inform you that your manuscript 'A versatile workflow to integrate RNA-seq genomic and transcriptomic data into mechanistic models of signaling pathways' has been provisionally accepted for publication in PLOS Computational Biology.

Best regards,

Mihaela Pertea

Software Editor

PLOS Computational Biology

Mihaela Pertea

Software Editor

PLOS Computational Biology

Reviewer's Responses to Questions

**Comments to the Authors:**

Reviewer #1: The manuscripts has been improved considerably and my original concerns have been addressed.

Reviewer #2: The authors have answered all previous comments. The improvements to the documentation and dockerization of the workflow make this a useful tool for the analysis of RNA-seq data.

**Have all data underlying the figures and results presented in the manuscript been provided?**

Reviewer #1: Yes

Reviewer #2: Yes

PLOS authors have the option to publish the peer review history of their article (what does this mean?). If published, this will include your full peer review and any attached files.

Reviewer #1: No

Reviewer #2: **Yes: **Alexander Lachmann

---

## [Editor Report · Acceptance letter]

7 Feb 2021

PCOMPBIOL-D-20-00902R1 

A versatile workflow to integrate RNA-seq genomic and transcriptomic data into mechanistic models of signaling pathways

Dear Dr Dopazo,

I am pleased to inform you that your manuscript has been formally accepted for publication in PLOS Computational Biology. Your manuscript is now with our production department and you will be notified of the publication date in due course.

With kind regards,

Alice Ellingham
